# Different Periampullary Types and Subtypes Leading to Different Perioperative Outcomes of Pancreatoduodenectomy: Reality and Not a Myth; An International Multicenter Cohort Study

**DOI:** 10.3390/cancers16050899

**Published:** 2024-02-23

**Authors:** Bas A. Uijterwijk, Daniël H. Lemmers, Giuseppe Kito Fusai, Bas Groot Koerkamp, Sharnice Koek, Alessandro Zerbi, Ernesto Sparrelid, Ugo Boggi, Misha Luyer, Benedetto Ielpo, Roberto Salvia, Brian K. P. Goh, Geert Kazemier, Bergthor Björnsson, Mario Serradilla-Martín, Michele Mazzola, Vasileios K. Mavroeidis, Santiago Sánchez-Cabús, Patrick Pessaux, Steven White, Adnan Alseidi, Raffaele Dalla Valle, Dimitris Korkolis, Louisa R. Bolm, Zahir Soonawalla, Keith J. Roberts, Miljana Vladimirov, Alessandro Mazzotta, Jorg Kleeff, Miguel Angel Suarez Muñoz, Marc G. Besselink, Mohammed Abu Hilal

**Affiliations:** 1Department of Surgery, Fondazione Poliambulanza, 25124 Brescia, Italy; basuijterwijk@live.nl (B.A.U.); d.h.lemmers@amsterdamumc.nl (D.H.L.); 2Department of Surgery, Amsterdam UMC, University of Amsterdam, 1012 Amsterdam, The Netherlands; m.g.besselink@amsterdamumc.nl; 3Cancer Center Amsterdam, 1081 Amsterdam, The Netherlands; 4Department of Surgery, Royal Free London NHS Foundation Trust, London NW3 2QG, UK; g.fusai@ucl.ac.uk; 5Erasmus MC, 3015 Rotterdam, The Netherlands; 6Department of Surgery, Fiona Stanley Hospital, Murdoch, WA 6150, Australia; sharnice.koek@health.wa.gov.au; 7Department of Pancreatic Surgery, IRCCS Humanitas Research Hospital, 20089 Rozzano, Italy; alessandro.zerbi@hunimed.eu; 8Division of Surgery, Department of Clinical Science, Intervention and Technology, Karolinska Institutet, Karolinska University Hospital, 171 64 Solna, Sweden; ernesto.sparrelid@ki.se; 9Department of Surgery, Pisa University Hospital, 56100 Pisa, Italy; ugo.boggi@unipi.it; 10Department of Surgery, Catharina Hospital Eindhoven, 5623 Eindhoven, The Netherlands; misha.luyer@catharinaziekenhuis.nl; 11Department of Surgery, Hospital del Mar, 08003 Barcelona, Spain; ielpo.b@gmail.com; 12Department of Surgery, University Hospital of Verona, 37126 Verona, Italy; roberto.salvia@univr.it; 13Department of Hepatopancreatobilliary and Transplant Surgery, National Cancer Centre, Singapore General Hospital, Singapore 168583, Singapore; bsgkp@hotmail.com; 14Surgery Academic Clinical Programme, Duke-National University of Singapore Medical School, Singapore 169857, Singapore; 15Department of Surgery, Amsterdam UMC, Location VUmc, 1007 Amsterdam, The Netherlands; g.kazemier@amsterdamumc.nl; 16Department of Surgery in Linköping, Department of Biomedical and Clinical Sciences, Linköping University, 581 83 Linköping, Sweden; bergthor.bjornsson@liu.se; 17Department of Surgery, Miguel Servet University Hospital, 50009 Zaragoza, Spain; marioserradilla@hotmail.com; 18Division of Oncologic and Mini-Invasive General Surgery, ASST Grande Ospedale Metropolitano Niguarda, 20162 Milan, Italy; micmazzola@gmail.com; 19Department of Academic Surgery, The Royal Marsden Hospital, London SW3 6JJ, UK; vasileios.mavroeidis@rmh.nhs.uk; 20Department of Hepatobiliary and Pancreatic Surgery, Oxford University Hospitals, NHS Foundation Trust, Oxford OX3 9DU, UK; zahir.soonawalla@ouh.nhs.uk; 21Department of Surgery, Hospital de Sant Pau, 08025 Barcelona, Spain; sanchez.cabus@gmail.com; 22Hepatobiliary and Pancreatic Surgical Unit, Nouvel Hôpital Civil (NHC), 67000 Strasbourg, France; patrick.pessaux@chru-strasbourg.fr; 23Department of Surgery, Newcastle Upon Tyne Hospitals, NHS Foundation Trust, Newcastle Upon Tyne NE3 3HD, UK; steven.white8@nhs.net; 24Department of Surgery, Virginia Mason, Seattle, WA 98101, USA; adnan.alseidi@ucsf.edu; 25Department of Surgery, University Hospital of Parma, 43126 Parma, Italy; raffaele.dallavalle@unipr.it; 26Department of Surgery, Hellenic Anticancer Hospital ‘Saint Savvas’, 115 22 Athens, Greece; dkorkolis_2000@yahoo.com; 27Department of Surgery, University Medical Center Schleswig-Holstein, Campus Lübeck, 23538 Lübeck, Germany; louisa.bolm@googlemail.com; 28Faculty of Medicine, University of Birmingham, Birmingham B15 2TT, UK; keith.roberts@uhb.nhs.uk; 29Department of Surgery Hospital Nuremberg, PMU Nürnberg, 90419 Nürnberg, Germany; miljana.vladimirov@gmail.com; 30Department of Digestive, Oncologic and Metabolic Surgery, Institut Mutualiste Montsouris, 75014 Paris, France; alex.mazzotta@gmail.com; 31Department of Surgery, Martin-Luther University Halle-Wittenberg, 06108 Halle (Saale), Germany; kleeff@gmx.de; 32Department of Surgery, University Hospital Virgen de la Victoria, 29010 Malaga, Spain; masuarez59@gmail.com

**Keywords:** non-pancreatic periampullary cancer, complications, pancreatoduodenectomy, tumor behavior

## Abstract

**Simple Summary:**

For cancer in the periampullary region, surgical resection with pancreatoduodenectomy remains the main curative treatment. Variations in prognosis suggest distinct growth patterns and tissue reactions, potentially influencing complications and perioperative mortality. This study aims to explore the impact of the type of periampullary adenocarcinoma on the perioperative hospital course.

**Abstract:**

This international multicenter cohort study included 30 centers. Patients with duodenal adenocarcinoma (DAC), intestinal-type (AmpIT) and pancreatobiliary-type (AmpPB) ampullary adenocarcinoma, distal cholangiocarcinoma (dCCA), and pancreatic ductal adenocarcinoma (PDAC) were included. The primary outcome was 30-day or in-hospital mortality, and secondary outcomes were major morbidity (Clavien-Dindo 3b≥), clinically relevant post-operative pancreatic fistula (CR-POPF), and length of hospital stay (LOS). Results: Overall, 3622 patients were included in the study (370 DAC, 811 AmpIT, 895 AmpPB, 1083 dCCA, and 463 PDAC). Mortality rates were comparable between DAC, AmpIT, AmpPB, and dCCA (ranging from 3.7% to 5.9%), while lower for PDAC (1.5%, *p* = 0.013). Major morbidity rate was the lowest in PDAC (4.4%) and the highest for DAC (19.9%, *p* < 0.001). The highest rates of CR-POPF were observed in DAC (27.3%), AmpIT (25.5%), and dCCA (27.6%), which were significantly higher compared to AmpPB (18.5%, *p* = 0.001) and PDAC (8.3%, *p* < 0.001). The shortest LOS was found in PDAC (11 d vs. 14–15 d, *p* < 0.001). Discussion: In conclusion, this study shows significant variations in perioperative mortality, post-operative complications, and hospital stay among different periampullary cancers, and between the ampullary subtypes. Further research should assess the biological characteristics and tissue reactions associated with each type of periampullary cancer, including subtypes, in order to improve patient management and personalized treatment.

## 1. Introduction

Periampullary adenocarcinoma is a common determinator for a diverse group of adenocarcinomas in and around the ampulla of Vater. In anatomic classification, periampullary adenocarcinoma includes pancreatic ductal adenocarcinoma (PDAC), distal cholangiocarcinoma (dCCA), duodenal adenocarcinoma (DAC), and ampullary adenocarcinoma [1,2,3].

Pancreatic ductal adenocarcinoma (PDAC) represents the most commonly diagnosed periampullary cancer, and it is associated with comparatively unfavorable prognoses following surgical resection [4]. The distinct origins of periampullary cancers result in varied reported survival rates, emphasizing the challenges in achieving favorable outcomes across different types.

Ampullary adenocarcinoma stands out among periampullary cancers due to its distinctive histomorphology and can be further categorized into histopathological subtypes, namely the intestinal, pancreaticobiliary, and mixed subtypes [5]. The intestinal subtype (AmpIT) shares histological similarities with DAC and small intestinal cancer, while the pancreaticobiliary subtype (AmpPB) exhibits histological resemblances to the distal bile duct and pancreatic duct epithelia.

Surgical resection with a pancreatoduodenectomy is the primary curative treatment for periampullary adenocarcinoma. However, variations in prognosis after surgery have been demonstrated [6,7]. These differences in prognosis suggest underlying differences in the growth pattern, which may lead to different surrounding tissue reactions, resulting in distinct challenges during the pancreatoduodenectomy, and may potentially contribute to variations in complications and post-operative mortality. Despite the presence of this hypothesis among surgeons, it has never been reported whether the periampullary cancer type affects mortality and post-operative complications during the initial hospital admission.

The objective of this study is to investigate the influence of periampullary adenocarcinoma type on the perioperative hospital course, specifically examining mortality rates, major morbidity, post-operative complications, and length of hospital stay.

## 2. Materials and Methods

### 2.1. Study Design

This study was a multicenter international observative cohort study including 30 centers (27 in Europe, one in the USA, one in Asia, one in Australia) of the international study group on non-pancreatic periampullary cancer (ISGACA; www.isgaca.com, accessed on 1 January 2020). The primary focus of the ISGACA consortium is to assess non-pancreatic periampullary cancers, defined as DAC, dCCA, and ampullar cancer (including AmpIT and AmpPB). A comparison between non-pancreatic periampullary cancer and pancreatic ductal adenocarcinoma (PDAC) was made, since this was considered of relevance during the perioperative course and this had not yet been compared. Therefore, data from patients who had undergone resection for PDAC were also gathered. These patients were recruited from five prominent ISGACA centers, including four in Europe and one in Australia, all within the same study period (see Appendix A). This study follows the STROBE (Strengthening the Reporting of Observational Studies in Epidemiology) guidelines, which are a set of international standards designed to enhance the transparency and completeness of reporting in observational studies. Following the STROBE guidelines signifies a commitment to rigorous and comprehensive reporting, ensuring that critical elements of study design, conduct, and analysis are clearly and thoroughly communicated [8]. The study was approved by the Ethical Committee of Brescia (number NP 5269–STUDIO NPPC 15 March 2022).

### 2.2. Patients

The inclusion criteria for this study comprised adult patients who underwent pancreatoduodenectomy and received a confirmed pathological diagnosis of DAC, AmpIT, AmpPB, dCCA, or PDAC within the time frame spanning from 2010 to 2021. Notably, patients with a mixed or hybrid subtype of ampullary carcinoma were deliberately excluded from this study due to variations in the definition and characteristics associated with these particular subtypes. This exclusion aimed to maintain clarity and consistency in the study population, focusing specifically on the defined and distinct histopathological subtypes of interest [9]. Patients who underwent surgery with palliative intent or following neoadjuvant chemotherapy were excluded from the study. Additionally, cases involving benign neoplasms, hybrid procedures, instances with missing primary outcome data, or those operated upon using alternative surgical techniques (such as total pancreatectomy, duodenum sparing pancreatectomy, and ampullectomy) were also excluded from the final analyses. This comprehensive exclusion criteria aimed to ensure a focused and consistent dataset for the study’s analysis (Appendix A). 

### 2.3. Data Collection and Definitions

Collected demographic data were sex, age (years), body mass index (BMI—Kg/m^2^), American Society of Anesthesiologists (ASA) classification [10], occurrence of vascular resection (both arterial and venous), minimally invasive/open approach, estimated perioperative blood loss (cc), operation time (minutes), and the 7th edition of the American Joint Committee on Cancer (AJCC) T and N staging. Collected outcome data were post-operative mortality, defined as “in-hospital” or “<30 days mortality”, major morbidity, defined as ≥Clavien-Dindo 3b, and the clinically relevant complications, which were clinically relevant post-operative pancreatic fistula (CR-POPF) [11], post-pancreatectomy hemorrhage (CR-PPH) [12], bile leakage (CR-BL) [13], and delayed gastric emptying (CR-DGE) [14], all defined as grade B and C. 

### 2.4. Surgical Techniques and Post-Operative Care

All patients underwent pancreatoduodenectomy. A pancreatoduodenectomy, also known as a Whipple procedure, is a complex surgical intervention involving the removal of the head of the pancreas, the duodenum, a portion of the common bile duct, the gallbladder, and sometimes a portion of the stomach [15,16]. Due to the pragmatic design of the study, no specific standards were provided for the surgical technique. Potential variations were pylorus-preserving or pylorus-resecting pancreatoduodenectomy and minimally invasive (MIPD) or open (OPD) pancreatoduodenectomy [17,18]. Participating centers followed local standard post-operative protocols aimed for effective patient recovery. The study design did not affect the post-operative care and there were no restrictions on blood tests, drain management, medication usage, or other co-interventions. However, participating centers are expected to adhere to consistent post-operative care for all tumor groups, following enhanced recovery principles that emphasize early mobilization and a gradual increase in oral intake based on patient preferences.

### 2.5. Statistical Analyses

The statistical analyses were conducted to compare the demographic and intraoperative characteristics among distinct histopathological subtypes, including PDAC, AmpIT, AmpPB, dCCA, and DAC. The statistical significance level was decided to be below 0.05, two sided. Descriptive statistics, such as means with standard deviation (SD) for normally distributed variables and medians with interquartile ranges (IQR) for non-normally distributed variables, were employed to outline and compare these characteristics across the specified subtypes. Categorical variables were presented in terms of frequencies and proportions. The Chi Square-test was utilized for the comparison of categorical data, providing insights into the distribution of specific characteristics among the different subtypes. Numerical data underwent evaluation through either Student-t test for normally distributed variables or the Mann–Whitney U test for non-normally distributed variables, allowing for a comprehensive comparison of quantitative aspects. In addition, both univariate and multivariate analyses were performed to assess the impact of various factors on mortality and major morbidity. The univariate analyses explored the individual effects of variables, while the multivariate analyses adjusted for potential confounding factors, providing a more nuanced understanding of the relationships within the PDAC, AmpIT, AmpPB, dCCA, and DAC groups.

## 3. Results

### 3.1. Demographics

Overall, 3622 patients were included, of which 370 were DAC, 811 were AmpIT, 895 were AmpPB, 1083 were dCCA, and 463 were PDAC. The selection of patients is displayed in Appendix A and the demographics of the cohorts is reported in Table 1. Most demographics were balanced between the groups, as reported in Table 1. In the dCCA cohort, there were fewer female patients (36%) compared to the other periampullary cancers (*p* < 0.001). The BMI ranged from 24.9 to 25.6. In the dCCA group, less patients are classified with a high ASA classification compared to the other groups (compared to AmpPB, 29.6 vs. 36.4%, *p* = 0.005). 

### 3.2. Intraoperative Outcomes

Vascular resection was most frequently performed in PDAC (26.4%), followed by dCCA (11.7%), AmpPB (2.1%), AmpIT (2.0%), and DAC (1.7%, *p* < 0.001, Table 1). The estimated perioperative blood loss was the lowest in AmpIT (350 cc) and the most in PDAC and dCCA (500 cc, *p* < 0.001). The operation time was the shortest in DAC (336 min) and the longest in dCCA (375 min, *p* < 0.001). 

### 3.3. Post-Operative Outcomes

The incidence of post-operative mortality was comparable between DAC, AmpIT, AmpPB, and dCCA (range 3.7–5.9%), while significantly lower in PDAC (1.5%, *p* = 0.043, Figure 1). Major morbidity was not significantly different between DAC, AmpIT, and AmpPB (69, 19.9%; 83, 14.4%; 108, 15.4%, respectively, *p* > 0.05); dCCA was significantly higher compared to AmpPB (18.6% vs. 15.4%, *p* = 0.002), and PDAC was significantly lower compared to dCCA (4.4% vs. 18.6%, *p* < 0.001). The highest incidence rates of CR-POPF were found in DAC (27.3%), AmpIT (25.5%), and dCCA (27.6%), and the lowest in PDAC (8.3%, compared to dCCA, *p* < 0.001), while AmpPB was found to be in the middle (18.5%, compared to Amp IT, *p* = 0.004, compared to dCCA, *p* = 0.001, Table 2). The length of hospital stay for patients with PDAC was 11 days, which was significantly shorter compared to 14–15 days for the other periampullary cancers (*p* < 0.001).

The other complications are assessed and reported in Appendix A. The incidence of CR-PPH could be significantly divided into high and low, with high in DAC (12.6%) and dCCA (11.1%), and low in AmpIT (8.2%), AmpPB (6.6%), and PDAC (6.2%). The incidence of CR-BL was comparable between DAC, AmpIT, AmpPB, and dCCA (range 5.9–7.0%), while significantly lower in PDAC (3.7%, *p* = 0.018, Appendix A). The incidence of CR-DGE was lowest in PDAC (1.9%), followed by dCCA (9.5%, *p* < 0.001), while DAC, AmpIT, and AmpPB show higher incidences (15.1%, 15.3%, and 14.3%, respectively). 

### 3.4. Multivariate Analyses 

The differences in mortality and major morbidity were adjusted for potential confounding factors using a multivariate model (see Table 2). For mortality, the variables PDAC, age, ASA, and N-stage demonstrated a significant effect (*p* = 0.018, *p* < 0.001, *p* < 0.001, *p* = 0.049, respectively) on mortality in univariate analyses. When combined in a multivariate model, PDAC (*p* = 0.021), age (*p* < 0.001), and ASA (*p* = 0.009) remained significant. For major morbidity, the variables PDAC (*p* < 0.001), age (*p* < 0.001), ASA (*p* < 0.001), N-stage (*p* < 0.001), and resection margin (*p* < 0.001) were significant in univariate analyses. In a multivariate model, PDAC (*p* = 0.025), age (*p* < 0.001), ASA (*p* < 0.001), and N-stage (*p* = 0.031) remained significant.

## 4. Discussion

This international multicenter retrospective cohort study is the first to report on the different perioperative outcomes across periampullary adenocarcinomas. PDAC appears to have the lowest mortality, lowest major morbidity, and the shortest hospital stay. DAC, AmpIT, and dCCA appear to have the highest CR-POPF rate, and DAC and dCCA have the longest hospital stay. Additionally, despite their close anatomical resemblance, the ampullary subtypes reveal variations in CR-POPF rates.

Numerous studies have examined the technical risk factors of the anastomosis technique associated with POPF (pancreaticojejunostomy vs. pancreaticogastrostomy, modified Blumberg vs. dunking anastomosis, type of suture) [19,20,21,22]. Some studies, including the updated alternative fistula risk score [23], have shown that the occurrence of pancreatic fistula is lower with a firmer pancreatic texture [24], and other studies have demonstrated a lower percentage of fistula in PDAC and more in dCCA and ampullary cancer [25]. Furthermore, for PDAC specifically, increased fibrosis, acinar atrophy, and chronic inflammation is found to be associated with worse survival outcomes [26]. Additionally, one study indicates a negative correlation between intratumoral necrosis and survival [27], whereas another study has found no significant relationship between stroma density and tumor progression or survival [28]. In this study, including a large patient cohort treated at various medical centers by different surgeons using distinct techniques, the primary factors influencing the incidence of POPF are intrinsic to the pancreas itself and the associated parenchymal and ductal changes linked to the disease. This study shows a lower percentage of CR-POPF in the “pancreas-related” malignancies (PDAC and AmpPB) and a higher percentage CR-POPF in the “intestinal- or bile duct-related” malignancies (DAC, AmpIT, and dCCA). The difference in CR-POPF between the subtypes of ampullary cancer is particularly notable. This study showed that AmpPB had a lower incidence of CR-POPF. The lower incidence of POPF in AmpPB and PDAC compared to DAC, AmpIT, and dCCA is likely due to the more favorable tissue characteristics of the pancreas in AmpPB and PDAC, suggesting a firmer texture. These results contribute valuable insights into the nuanced factors influencing CR-POPF outcomes in distinct malignancies. 

Despite the anatomical resemblance of the ampullary subtypes, AmpPB exhibits less CR-POPF when compared to AmpIT. This suggests that, in addition to the periampullary tumor type, outcomes are also influenced by the histopathological subtype of ampullary adenocarcinoma. Less CR-POPF in AmpPB suggests more biological processes such as increased fibrosis, acinar atrophy, or inflammation leading to more favorable tumor characteristics for the most optimal pancreaticojejunostomy. Although it should be further investigated what these specific factors are, this outcome even further underscored the differences between the ampullary subtypes in tumor biology. 

It was observed that the length of hospital stay for PDAC was 3–4 days shorter, and mortality was 2.2–4.4% lower compared to the other periampullary cancers (*p* < 0.05). This is anticipated to be associated with the lower incidence of complications in PDAC cases. Further studies are warranted to establish direct correlations between tumor biology, tissue fibrosis, acinar atrophy, and post-operative complications. However, it is imperative to bear in mind that patients with certain tumor pathologies are at a higher risk, necessitating enhanced diagnostics for early complication detection during post-operative care.

Several prevalent beliefs align with our findings in the context of pancreatoduodenectomy and its associated anastomoses. Pancreatoduodenectomy involves three primary anastomoses: pancreatojejunostomy (or less frequently, gastrojejunostomy), hepatojejunostomy, and gastro/duodeno-jejunostomy (depending on pylorus preservation or resection). The convention dictates that a dilated bile duct or pancreatic duct during pancreatoduodenectomy is generally considered beneficial for anastomosis, with ampullary carcinoma potentially affecting both the pancreatic and bile ducts, while bile duct cancer may specifically involve a dilated bile duct. Moreover, a firmer pancreas is thought to be beneficial for anastomosis creation [22]. Although PDAC originates from pancreatic tissue, contributing to a lower incidence of POPF, interestingly, ampullary adenocarcinoma related to pancreatic tissue, particularly AmpPB, exhibits a lower fistula rate compared to dCCA, DAC, and AmpIT. Lastly, while there is a surgical concern regarding a dilated stomach due to duodenal obstruction, impacting anastomosis for DAC, there is currently no available data to substantiate this claim. These observations emphasize the complex interplay between anatomical considerations and the diverse periampullary cancer subtypes during pancreatoduodenectomy.

To date, there is only marginal research on molecular classification, yet its potential significance in enhancing the categorization of periampullary cancers cannot be overlooked. A molecular classification approach, utilizing specific tumor markers such as CK7, CK20, MUC1, MUC2, and CDX2, and genes like KRAS, TP53, APC, and PIK3CA, could prove pivotal in refining the classification methodology [29,30]. However, it is important to note that, as of now, there is no standardized technique for the implementation of tumor markers. The development of future studies is crucial to address this gap in the knowledge and to establish consistent methodologies.

This study has certain limitations that need to be taken into account when interpreting the results. First, the assessment of pancreas texture and duct size was not conducted, which could potentially predict the occurrence of complications [22]. Future studies should address this aspect to gain a better understanding of its influence. Second, the incidence of DGE in PDAC may have been underestimated, considering the expected higher occurrence in this tumor type [31]. It is important to interpret this finding cautiously and further investigate this in future studies. Third, vascular and arterial resection are collected as one variable; however, future studies should considerer collecting both venous and arterial resection separately as it can affect outcomes [32,33]. Fourth, it is important to acknowledge that the cases included in this study are not consecutive. The primary focus of the initiating study group lies on non-pancreatic periampullary cancer. Consequently, to facilitate a comprehensive comparison, separate data collection was necessary for PDAC cases. To mitigate potential bias, these PDAC cases were gathered from five prominent centers. Nonetheless, a certain level of bias may still persist in the separately collected PDAC cases. Therefore, it is important to carefully consider and account for this potential bias in our analysis. Fifth, re-admission continues to be a crucial metric for assessing treatment quality. Unfortunately, these data were not accessible for this study. It is imperative that future research should include the evaluation of re-admission rates for a more comprehensive understanding of treatment outcomes. Sixth, the classification of the five different periampullary cancers was based on WHO guidelines and local protocols, as there were no internationally validated and standardized definitions. This reliance on varied criteria may introduce minor differences in classification between centers, highlighting the need for an international consensus on defining periampullary tumors. Additionally, the mixed subtype of ampullary cancer, characterized by features of both intestinal and pancreatobiliary subtypes, poses challenges in differentiation. While this study excludes the mixed subtype for clarity, its existence should be acknowledged. Therefore, efforts toward establishing consensus on the definition and classification of the mixed subtype are essential for future research and clinical understanding.

While this study may not provide a definitive solution on the topic, it does possess several strengths which should be acknowledged. First, it is the first study to examine and statistically document the correlation between periampullary adenocarcinoma types and complications, providing a foundation for future research into this topic. Second, the findings of this study provide evidence-based support to clinicians in their daily decision-making, regarding the assessment and management of post-operative patients and, thus, paying more attention with a low threshold for the radiological assessment of high-risk patients with potential complications, even at the slightest indication of concern. Third, the study’s international design enhances the global applicability and relevance of its results, and it limits any potential cultural biases by including patients from various countries and continents operated on by different surgeons. Lastly, the comprehensive assessment of ampullary subtypes supports their clinical significance and supports the need for a separate evaluation in future studies [7]. 

## 5. Conclusions

In conclusion, this study outlines the differences in perioperative mortality, post-operative complications, and hospital stay among different periampullary adenocarcinomas and ampullary subtypes. Understanding these differences is crucial for understanding the underlying biology of each tumor type. Future research should further investigate the biological impact of periampullary adenocarcinomas on the surrounding tissue, and healthcare professionals should consider the type of periampullary adenocarcinoma when providing tailored treatment plans for these patients.

## Figures and Tables

**Figure 1 cancers-16-00899-f001:**
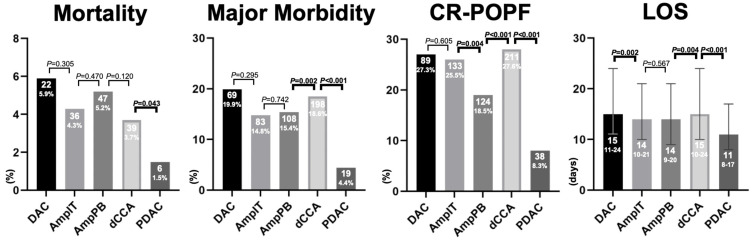
Primary and secondary outcomes per periampullary adenocarcinoma. Abbreviations: CR-POPF, clinically relevant post operative pancreatic fistula; LOS, length of hospital stay in days; DAC, duodenal adenocarcinoma; AmpIT, ampullary adenocarcinoma subtype intestinal; AmpPB, ampullary adenocarcinoma subtype pancreatobiliary; PDAC, pancreatic ductal adenocarcinoma; Morality, 30-day or in-hospital mortality; Major Morbidity, ≥Clavien-Dindo 3b; LOS, length of hospital stay.

**Table 1 cancers-16-00899-t001:** Demographics and intraoperative outcomes.

		DAC	*p* Value (DAC vs. AmpIT)	AmpIT	*p* Value (AmpIT vs. AmpPB)	AmpPB	*p* Value (AmpPB vs. dCCA)	dCCA	*p* Value (dCCA vs. PDAC)	PDAC	*p* Value Total
** *n* **		370		811		895		1083		463	
** *Sex (F/M), n (%)* **		153 (41.4)	*0.660*	348 (42.9)	*0.589*	371 (41.5)	** *0.010* **	387 (35.7)	** *<0.001* **	212 (45.8)	** *0.001* **
** *Age §* **		67 [60, 73]	*0.067*	68 [61, 75]	*0.287*	69 [61, 75]	*0.111*	68 [61, 74]	*0.594*	68 [61, 74]	*0.068*
** *BMI §* **		25 [23, 28]	*0.630*	25 [23, 28]	*0.170*	25 [23, 28]	** *0.016* **	25 [23, 28]	** *0.005* **	26 [23, 29]	** *0.023* **
** *ASA, n (%)* **	** *1&2* **	247 (67.3)	*0.452*	486 (64.1)	*0.524*	509 (63.6)	** *0.005* **	725 (70.4)	*0.136*	300 (64.9)	*0.091*
	** *3&4* **	120 (32.7)		272 (35.9)		291 (36.4)		305 (29.6)		162 (35.1)	
**Vascular res. *n*, (%)**		6 (1.7)	*0.917*	14 (2.0)	*1000*	16 (2.1)	** *<0.001* **	120 (11.7)	** *<0.001* **	120 (26.4)	** *<0.001* **
**MIS, *n* (%)**		32 (8.6)	*0.284*	62 (11.0)	** *0.009* **	41 (6.6)	** *0.003* **	120 (11.1)	** *<0.001* **	15 (3.2)	** *<0.001* **
**Blood loss, cc §**		450 [245, 800]	*0.112*	350 [200, 700]	*0.623*	400 [200, 650]	** *0.001* **	500 [289, 700]	** *<0.001* **	500 [300, 900]	** *<0.001* **
**Op. time, min §**		336 [260, 430]	*0.147*	360 [270, 440]	** *0.047* **	364 [300, 447]	*0.148*	375 [300, 473]	*0.130*	360 [310, 420]	** *<0.001* **

Abbreviations: DAC, duodenal adenocarcinoma; AmpIT, ampullary adenocarcinoma subtype intestinal; AmpPB, ampullary adenocarcinoma subtype pancreatobiliary; PDAC, pancreatic ductal adenocarcinoma; F, female; M, male; ASA, American Society of Anesthesiologists; BMI, body mass index; Vascular res., vascular resection; MIS, minimally invasive surgery approach; Op. time, operation time; *p*, *p*-value; §, (median [IQR]); bold values correspond with <0.05 significance.

**Table 2 cancers-16-00899-t002:** Univariate and multivariate logistic regression model for (**A**) 30-day or in-hospital mortality and (**B**) major morbidity based on the different histopathological periampullary adenocarcinoma subtypes.

**(A) Mortality** **(30 d/in-Hospital)**	**Univariate**				**Multivariate**			
	**Coeff.**	**Std. Error**	**Z-Value**	***p*-Value**	**Coeff.**	**Std. Error**	**Z-Value**	***p*-Value**
AmpPB vs. AmpIT	0.251	0.247	1.014	*0.310*	0.260	0.258	1.005	*0.315*
dCCA vs. AmpIT	−0.132	0.250	−0.529	*0.597*	−0.158	0.261	−0.607	*0.544*
DAC vs. AmpIT	0.343	0.290	1.182	*0.237*	0.329	0.298	1.103	*0.270*
PDAC vs. AmpIT	−1.071	0.453	−2.366	** *0.018* **	−1.052	0.457	−2.301	** *0.021* **
Age (years)	0.040	0.010	4.086	** *<0.001* **	0.037	0.010	3.535	** *<0.001* **
ASA 3/4	0.619	0.179	3.458	** *<0.001* **	0.485	0.187	2.597	** *0.009* **
T stage 3/4	−0.050	0.195	−0.258	*0.797*				
N stage 1/2	−0.350	0.178	−1.972	** *0.049* **	−0.296	0.186	−1.592	*0.111*
Resection margin	−0.270	0.220	−1.227	*0.220*				
Perineural invasion	−0.148	0.185	−0.799	*0.424*				
Lymphovascular invasion	0.041	0.186	0.222	*0.824*				
MIS	0.069	0.310	0.222	*0.824*				
**(B) Major Morbidity** **(CD 3b≥)**	**Univariate**				**Multivariate**			
	**Coeff.**	**Std. Error**	**Z-Value**	***p*-Value**	**Coeff.**	**Std. Error**	**Z-Value**	***p*-Value**
AmpPB vs. AmpIT	0.088	0.153	0.575	*0.566*	0.218	0.164	1.327	*0.185*
dCCA vs. AmpIT	0.271	0.139	1.955	*0.051*	0.260	0.153	1.692	*0.091*
DAC vs. AmpIT	0.323	0.174	1.855	*0.064*	0.406	0.181	2.242	** *0.025* **
PDAC vs. AmpIT	−1.169	0.247	−4.729	** *<0.001* **	−1.020	0.261	−3.912	** *<0.001* **
Age (years)	0.018	0.005	3.535	** *<0.001* **	0.013	0.006	2.361	** *0.018* **
ASA 3/4	0.439	0.103	4.267	** *<0.001* **	0.473	0.111	4.279	** *<0.001* **
T stage 3/4	−0.064	0.113	−0.564	*0.573*				
N stage 1/2	−0.343	0.102	−3.352	** *<0.001* **	−0.240	0.111	−2.157	** *0.031* **
Resection margin	−0.471	0.129	−3.645	** *<0.001* **	−0.262	0.143	−1.834	*0.067*
Perineural invasion	−0.050	0.106	−0.473	*0.636*				
Lymphovascular invasion	−0.123	0.105	−1.17	*0.242*				
MIS	0.298	0.163	1.835	*0.067*				

Abbreviations: AmpPB, ampullary pancreatobiliary-type adenocarcinoma; AmpIT, ampullary intestinal-type adenocarcinoma; dCCA, distal cholangiocarcinoma; DAC, duodenal adenocarcinoma; PDAC, pancreatic ductal adenocarcinoma; ASA, American Society of Anesthesiologists; T stage, tumor stage; N stage, lymph node stage; MIS, minimally invasive surgery.

## Data Availability

The original contributions presented in the study are included in the article, further inquiries can be directed to the corresponding author.

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
