# Peer review of "Different Periampullary Types and Subtypes Leading to Different Perioperative Outcomes of Pancreatoduodenectomy: Reality and Not a Myth; An International Multicenter Cohort Study"

_cancers, 2024, doi:10.3390/cancers16050899_

Round 1
Reviewer 1 Report
Comments and Suggestions for Authors
1. Data displacement need to be modified. for example, in Table 1, then how about DAC vs Amp PB, DAC vs dCCA.... Auhtor may be benefit from special statistician consultations.
2. Figure 1.
Mortality :consider list each p-value comparing with PDAC.
Major morbidity: no need for p-value between DAC vs. Amp/ DAC vs. PDAC?
CR-POPF: no need for p-value for DAC vs Amp PB/ DAC vs. PDAC...?
3. Table 3: It would be better author need to consider texture of the pancras, p-duct size, or BMI... and try to find out why individual diagnostic entities resulted in different postoperative morbidity and mortality.
4. Overall, very interesting study, but it would be better author need to further investigate, for example, why DAC was found to be associated with higher mortality, higher major morbidity, and CR-POPF... Clinico-pathological diagnosis correlation such as texture of the pancreas, p-duct size, BMI, and hospital voulme... might be one of the clues to solve this issue.
Comments on the Quality of English Languagevery a few minor edititing may be required.
Author Response
We want to thank the reviewer for his/her valuable time and effort. We have replied on the comments and suggestions and provided a point-by-point respons.
Comments and Suggestions for Authors
1. Data displacement need to be modified. for example, in Table 1, then how about DAC vs Amp PB, DAC vs dCCA.... Author may be benefit from special statistician consultations. Figure 1.
Mortality: consider list each p-value comparing with PDAC.
Major morbidity: no need for p-value between DAC vs. Amp/ DAC vs. PDAC?
CR-POPF: no need for p-value for DAC vs Amp PB/ DAC vs. PDAC...?
We want to express our gratitude to the reviewer for taking the time and effort to critically review our manuscript. We agree that in figure 1, all p-values are essential to report. We did not want to overstimulate the reader with too many p-values but indeed, if the p-value is >0.05, this is something the reader wants to read instead of believe the authors on it. Therefore, we added all p-values in the figure with bold representing significant values. We also want to thank the reviewer for the suggestion to compare all values with PDAC. However, primary aim was to assess them all. To prevent multiple testing we were required to make a decision. We again discussed this with all authors and we believe that when all are compared to PDAC, all are significant but that does not say anything because most of us know that non-pancreatic periampullary tumors are clearly different from PDAC, but how do they relate to each-other. Therefore, the order we made was based on the differences we know from literature, and that is long-term prognosis. We used studies as De Jong et al. (DOI: https://doi.org/10.1016/j.hpb.2022.01.009) and He et al. (DOI:
https://doi.org/10.1186/s12885-018-4240-x), to give us the vision that the order from best to worst prognosis, including ampullary subtypes, is DAC, AmpIT, AmpPB, dCCA, PDAC (based on current literature). This was the order taken and incorporated to demonstrate most relevant differences. For more specific sub questions, we need to perform sub studies, which can definitely follow if this study raises specific questions, such as focusing on bile leakage for dCCA vs PDAC or bile duct dilatation among dCCA, ampIT or ampPB. We are curious what the reviewer thinks of this.
3. Table 3: It would be better author need to consider texture of the pancreas, p-duct size, or BMI... and try to find out why individual diagnostic entities resulted in different postoperative morbidity and mortality.
Firstly, we want to thank the reviewer for this valuable insight and we fully agree. Unfortunately, factors such as texture of the pancreas and pancreatic duct size were not included during the start of the study. This has several reasons. First, since we know that the differences, we expected to found were very subtle, we were required to have a very long study period in order to include enough patients to see small differences. Second, due to international approach, applied in order to make the study applicable not only in our countries but the results could be translated to all centers globally. However, this international multicenter approach comes with intercontinental bias which requires to collect even more patients to compensate slight cultural differences. Therefore, study inclusion started in 2010 already. Back then, pancreatic texture and duct size was not yet on the radar. We tried to retrospectively include these values; however, this was not possible because this was simply not recorded in patient files back then.
However, we acknowledge the importance of these variables in the first limitation mentioned (line number 322-324). This is the first and definitely the largest study that demonstrate differences and advises future studies how to proceed on this topic. In this study we also stress that future studies assessing clinical course after PD also includes duct size and pancreatic texture.
4. Overall, very interesting study, but it would be better author need to further investigate, for example, why DAC was found to be associated with higher mortality, higher major morbidity, and CR-POPF... Clinico-pathological diagnosis correlation such as texture of the pancreas, p-duct size, BMI, and hospital voulme... might be one of the clues to solve this issue.
We want to express our gratitude for the compliment and for his/her valuable suggestions. We couldn’t agree more and we cannot wait to further investigate the topic. We feel that the reviewer is working in a large tertiary progressive center and is not surprised by the differences found and wants to know more. However, globally are many surgeons that believe that no matter what type of tumor it is in the head of the pancreas, the Whipple/PD will have the same impact and that after surgery, the tumor type starts to play a role. The primary aim of this study was to demonstrate that this is incorrect, the type of periampullary cancer you are dealing with does play a role in the perioperative care and knowing the differences could be in your advantage (such as lowering thresholds for post-op CAT scan or prolonged ICU/PACU care). The question why is extremely challenging and can’t be answered by one study. We can share our hypotheses and motivate other research teams to start assessing these differences with new study designs. We will not stop with this study and we will continue to delve deeper into the details per tumor type to eventually find the answers. Every study will enlighten a new detail regarding the tumor behavior which influences the surrounding tissue, and knowing them enables us to improve surgical techniques accordingly.
Reviewer 2 Report
Comments and Suggestions for Authors
Bas A. Uijterwijk and co-authors used the International Multicenter Cohort Study to describe the Different Periampullary Types and Subtypes leading to different Perioperative Outcomes of Pancreatoduodenectomy. Despite the promise, the manuscript has several important flaws that need to be addressed before publication:
1. In the study design authors included 30 centers but the patients were recruited from only five centers. Why?
2. Describe a little bit about NPPC.
3. Is there any specific region that the authors added only Females in Table 1?
4. Authors need to compare the statistics between each group.
5. CR-POPF missing in Table 2.
6. The p-value of CR-PORF and Mortality does not match the in-text figure 1.
7. The incidence of CR-BL in Supp Table S1 not in Table 2
8. Aline properly table 2.
9. In Table 2 PDAC vs AmpIT, the p-value is 0.018 but in text 0.080.
10. In Supplementary Figure S1:
The authors used full stops instead of commas in most of the areas.
The final database (n= 3622) does not match with each subtype.
Ampullary adenocarcinoma intestinal subtype repeated.
The numbers in DAC and PDAC are the same.
Comments on the Quality of English LanguageSome of the sentences are not completed scientifically.
Author Response
Comments and Suggestions for Authors
Bas A. Uijterwijk and co-authors used the International Multicenter Cohort Study to describe the Different Periampullary Types and Subtypes leading to different Perioperative Outcomes of Pancreatoduodenectomy. Despite the promise, the manuscript has several important flaws that need to be addressed before publication:
1. In the study design authors included 30 centers but the patients were recruited from only five centers. Why?
We would like to express our gratitude to the reviewer for taking the time to review our work and provide valuable feedback. Patients with ampullary, duodenal and distal bile duct adenocarcinoma were selected in all 30 participating centers. Since the data collection originally consisted of patients with non-pancreatic periampullary cancers, patients with PDAC, were collected in a later phase in five prominent centers. Since we did not report this clearly, and therefore we added a clearer explanation in the method sections on line numbers 106-113
2. Describe a little bit about NPPC.
In line number 107-108 we added the definition of non-pancreatic periampullary cancers (NPPC). However, since we believe the differences among NPPC are clinically relevant and therefore, the collective definition NPPC should be avoided. In the introduction, line number 92, we replaced NPPC with “periampullary adenocarcinoma”. This way it is clearer that this sentence is meant for AmpPB, AmpIT, DAC, dCCA and PDAC.
3. Is there any specific region that the authors added only Females in Table 1?
We sincerely thank the reviewer for his/her thoughtful feedback. We can now see how this could be interpreted on multiple ways. The number represents the number of female compared to man; the percentage is the percentage female / male participants. We agree that this is not totally clear the way we reported it and therefore, we changed the table and added this in the table legend in line number 201.
4. Authors need to compare the statistics between each group.
We appreciate the valuable suggestion from the author. Extensive discussions took place concerning the comparisons made. The primary goal was to comprehensively assess all periampullary tumors. To address the concern of multiple testing, a decision had to be made. After further deliberation with all authors, the consensus was that comparing each tumor type to PDAC individually would yield significance due to the well-known differences between non-pancreatic periampullary tumors and PDAC. However, the critical question was how these non-pancreatic periampullary tumors related to each other. Therefore, the established order (DAC, AmpIT, AmpPB, dCCA, PDAC) was based on long-term prognosis differences drawn from relevant literature, including studies by De Jong et al. (DOI: https://doi.org/10.1016/j.hpb.2022.01.009) and He et al. (DOI: https://doi.org/10.1186/s12885-018-4240-x). This order was chosen to highlight the most significant distinctions. For more specific inquiries, such as bile leakage in dCCA vs. PDAC or bile duct dilatation among dCCA, AmpIT, or AmpPB, dedicated sub-studies will be essential. We welcome the reviewer's thoughts on this approach.
5. CR-POPF missing in Table 2.
Thank you for your keen observation. We acknowledge that LOS (Length of Stay) was not outlined in the figure legend as well. This information has been incorporated into line 212 for clarity.
6. The p-value of CR-PORF and Mortality does not match the in-text figure 1.
Thank you for your attention to detail. The identified issue has been corrected, which contributes to an improved quality of the manuscript.
7. The incidence of CR-BL in Supp Table S1 not in Table 2
Thank you for bringing this to our attention. The error has been rectified.
8. Aline properly table 2.
We want to thank the reviewer for his/her sharp observation. We carefully aligned the tables throughout the manuscript.
9. In Table 2 PDAC vs AmpIT, the p-value is 0.018 but in text 0.080.
Thank you for bringing this to our attention. We appreciate the detailed review, allowing us to enhance the manuscript. The error has been corrected.
10. In Supplementary Figure S12:
The authors used full stops instead of commas in most of the areas.
We want to thank the reviewer for bringing this to our attention. In supplementary figure S1, line number 477, we replaced the full stops to indicate a thousand. Regarding supplementary table S1 in the submission, there was not a table legend included in the transition to the correct layout of the journal. Herby, we added for both supplementary figure 1 and supplementary table 1 the legend with abbreviations and explanations. Regarding the comma in the length of hospital stay, this represents the interquartile range. The following is added: “§, (median [IQR minimum, IQR maximum])”. The sign for a decimal point remains a point.
The final database (n= 3622) does not match with each subtype.
Ampullary adenocarcinoma intestinal subtype repeated.
The numbers in DAC and PDAC are the same.
Thank you for identifying the error. We have found the error in reporting while the calculations were found to be correct. Due to DAC and PDAC were mistakenly reported both 463, while this is the number for PDAC only. DAC should be 370. Subsequently, the total number of 3622 was calculated with the correct numbers and is therefore still correct. We corrected the error in the text.
Reviewer 3 Report
Comments and Suggestions for Authors
A very nice paper on tumours in a specific localization. The rationale is solid, with a high number of samples, and the conclusions are solid
English language is fine and references are updated
A minor improvement to the manuscript would be the correlation of the data with some molecular classifications, translating the clinical and histological findings to the next level, especially for ampullary cancer, which may explain the differences between the histological subtypes. Some papers have already addressed this subject (https://www.nature.com/articles/s41416-019-0415-8; https://www.sciencedirect.com/science/article/abs/pii/S0898656822002236), but inserting a paragraph would add quality to the discussion
Author Response
Comments and Suggestions for Authors
A very nice paper on tumours in a specific localization. The rationale is solid, with a high number of samples, and the conclusions are solid
English language is fine and references are updated
A minor improvement to the manuscript would be the correlation of the data with some molecular classifications, translating the clinical and histological findings to the next level, especially for ampullary cancer, which may explain the differences between the histological subtypes. Some papers have already addressed this subject (https://www.nature.com/articles/s41416-019-0415-8; https://www.sciencedirect.com/science/article/abs/pii/S0898656822002236), but inserting a paragraph would add quality to the discussion
Thank you for your valuable suggestion. We acknowledge the potential added insight that molecular classifications could bring to our study, especially in elucidating the differences between histological subtypes, particularly in ampullary cancer. We recognize the importance of translating our clinical and histological findings to a molecular level. While existing studies, have delved into this subject, we appreciate the merit of incorporating a paragraph addressing molecular classifications in our discussion. This addition will undoubtedly enhance the quality of our manuscript by providing a more comprehensive perspective on the underlying molecular mechanisms associated with the observed histological variations. We have added a paragraph in de discussion to elaborate on this topic in line numbers 313-320.
Reviewer 4 Report
Comments and Suggestions for Authors
Dear authors,
Congratulations, this is excellent original article.
This paper is suitable for publication in this journal.
Best Regards
Author Response
Comments and Suggestions for Authors
Dear authors,
Congratulations, this is excellent original article.
This paper is suitable for publication in this journal.
We express our sincere gratitude to the reviewer for dedicating time and effort to evaluate our manuscript. We appreciate the valuable feedback and the trust placed in our work.